# Clinical management of community-acquired meningitis in adults in the UK and Ireland in 2017: a retrospective cohort study on behalf of the National Infection Trainees Collaborative for Audit and Research (NITCAR)

Jayne Ellis,[1] David Harvey,[2] Sylviane Defres,[3,4] Arjun Chandna,[5] Eloisa MacLachlan,[6,7] Tom Solomon,[3,8] Robert S Heyderman [ID],[1] Fiona McGill [ID],[3,9] on behalf of the National Audit of Meningitis Management (NAMM) group

For numbered affiliations see end of article.

**Correspondence to**
Dr Fiona McGill;
f.mcgill@nhs.net

## ABSTRACT

**Objectives** To assess practice in the care of adults with suspected community-acquired bacterial meningitis in the UK and Ireland.

**Design** Retrospective cohort study.

**Setting** 64 UK and Irish hospitals.

**Participants** 1471 adults with community-acquired meningitis of any aetiology in 2017.

**Results** None of the audit standards, from the 2016 UK Joint Specialists Societies guideline on diagnosis and management of meningitis, were met in all cases. With respect to 20 of 30 assessed standards, clinical management provided for patients was in line with recommendations in less than 50% of cases. 45% of patients had blood cultures taken within an hour of admission, 0.5% had a lumbar puncture within 1 hour, 26% within 8 hours. 28% had bacterial molecular diagnostic tests on cerebrospinal fluid. Median time to first dose of antibiotics was 3.2 hours (IQR 1.3–9.2). 80% received empirical parenteral cephalosporins. 55% ≥60 years and 31% of immunocompromised patients received anti-*Listeria* antibiotics. 21% received steroids. Of the 1471 patients, 20% had confirmed bacterial meningitis. Among those with bacterial meningitis, pneumococcal aetiology, admission to intensive care and initial Glasgow Coma Scale Score less than 14 were associated with in-hospital mortality (adjusted OR (aOR) 2.08, 95% CI 0.96 to 4.48; aOR 4.28, 95% CI 1.81 to 10.1; aOR 2.90, 95% CI 1.26 to 6.71, respectively). Dexamethasone therapy was weakly associated with a reduction in mortality in both those with proven bacterial meningitis (aOR 0.57, 95% CI 0.28 to 1.17) and with pneumococcal meningitis (aOR 0.47, 95% CI 0.20 to 1.10).

**Conclusion** This study demonstrates that clinical care for patients with meningitis in the UK is not in line with current evidence-based national guidelines. Diagnostics and therapeutics should be targeted for quality improvement strategies. Work should be done to improve the impact of guidelines, understand why they are not followed

and, once published, ensure they translate into changed practice.

## STRENGTHS AND LIMITATIONS OF THIS STUDY

⇒ To our knowledge, this is the largest national study of the management of meningitis in the UK published to date.
⇒ The study includes all suspected community-acquired bacterial meningitis, allowing assessment of early clinical care prior to an aetiological diagnosis being made.
⇒ The study is widely translatable and representative of practice within the UK and Ireland.
⇒ The study is limited by its retrospective design, which brings associated recall bias and some missing data.
⇒ The study may also be limited by the self-selection of the sites included.

## INTRODUCTION

Acute bacterial meningitis is a medical emergency associated with considerable death and disability in the UK.[1] Successful immunisation programmes targeting *Haemophilus influenzae* type b, *Streptococcus pneumoniae* and *Neisseria meningitidis* means that community-acquired bacterial meningitis, particularly in children and adolescents, is now relatively rare.[2] The incidence of bacterial meningitis in adults in England is estimated to be approximately 1–1.25 per 100 000 population overall, exceeding 9 per 100 000 in people over 70 years.[2 3]

Early recognition of meningitis, appropriate investigation and treatment saves

lives.[4][5] It is essential that front-line clinicians, who may not encounter meningitis very often, are vigilant and have a high index of suspicion to minimise poor outcomes. To help staff who are seeing patients with suspected meningitis, the UK guidelines on the diagnosis and management of acute meningitis and meningococcal sepsis in immunocompetent adults were published in 2016.[6] The guidelines provide readily accessible, comprehensive, evidenced-based recommendations. Previous studies show that clinical care delivered in the UK is frequently non-adherent to guidelines.[7][8] A more recent UK study highlighted a large amount of inappropriate brain imaging prior to lumbar punctures (LPs) and long delays in performing LPs.[3][9] Inadequate use of molecular diagnostics and HIV testing have also been highlighted as areas for improvement.[3] The increasing risk of multidrug resistant bacteria, an ageing population susceptible to a wider variety of bacteria (eg, *Listeria monocytogenes, Escherichia coli* and *Klebsiella pneumoniae)*[2] and a greater appreciation that viruses are common causes of meningitis,[10][11] makes diagnostics essential. Reports from outside the UK have shown improvements in outcomes following guideline publication and implementation.[12] We carried out a retrospective observational study with the dual aims of (1) assessing current clinical practice regarding diagnosis and management of adult patients with suspected community-acquired bacterial meningitis and (2) to identify areas for improvement.

## METHODS

Hospitals in the UK were invited to take part in this study via the National Infection Trainees Collaborative for Audit and Research network, the UK Meningitis study network, the British Infection Association and through personal contacts. Eligible patients were identified via hospital coding data, laboratory data or a combination of both. Data from patients aged 16 or over who presented with suspected acute community-acquired bacterial meningitis during 2017 were eligible for screening. Patients who met our case definition for confirmed acute meningitis, regardless of aetiology, were eligible for inclusion (box 1). Definitions are as previously published.[3] Many interventions are performed prior to knowing the diagnosis, therefore, we included all meningitis in the analysis, including viral and those in whom no pathogen was identified. This allowed us to assess the entire clinical pathway of patients presenting with possible bacterial meningitis, although some would be ultimately diagnosed with a different aetiology.

Standards indicative of good practice were taken from the 2016 UK Joint Specialists Societies guideline on the diagnosis and management of meningitis and meningococcal sepsis in immunocompetent adults, and the Standards in Microbiological Investigations on the processing of cerebrospinal fluid (CSF) (B27).[6][13] For each standard, the number of patients as a proportion of the total cohort who received clinical care in line with the standard is

> **Box 1  Inclusion and exclusion criteria for cases of meningitis**
>
> **A meningitis case was defined as:**
> ⇒ Patients with a cerebrospinal fluid (CSF) white cell count>$4\times10^6$ cells/L (regardless of whether a pathogen was identified or not) and a clinical suspicion of meningitis at the time OR
> ⇒ In the case of bacterial meningitis, symptoms and signs of meningitis with a significant pathogen in the CSF (culture or PCR) or blood regardless of CSF leucocyte count
>
> **Patients with the following diagnoses were excluded:**
> ⇒ Cryptococcal meningitis
> ⇒ Tuberculous meningitis
> ⇒ Nosocomial meningitis (defined as meningitis that occurs during a hospital admission or within 30 days of discharge or meningitis associated with indwelling devices in the central nervous system)
> ⇒ Encephalitis (defined as altered consciousness for >24 with no other cause found and two or more of the following signs: fever or history of fever (≥38°C) during the current illness; seizures or focal neurological signs (with evidence of brain parenchyma involvement); CSF pleocytosis (>$4\times10^6$ cells/L); Electroencephalogram suggesting encephalitis; and neuroimaging suggestive of encephalitis).

reported. A second adjusted analysis taking account of missing data is also reported, whereby the number of patients as a proportion of the cohort with available data who received clinical care in line with the standard was reported.

Data were collected using electronic case report forms on REDcap, a password-protected central web-based database system. All microbiological diagnostic procedures were performed at the local hospital laboratory for each participating site using locally approved procedures. All data were anonymised and recorded under a unique participant identification number.

### Statistical analyses

Descriptive statistics were used to summarise data. Categorical data were summarised using counts and percentages. Denominators presented are based on available data, where incomplete case records were submitted by contributing sites. For continuous variables, means and ranges or medians and IQRs are presented depending on the distribution of the data. Categorical data were analysed using $\chi^2$ or Fisher's exact test. Continuous data were analysed using t-tests, Mann-Whitney U or Kruskal-Wallis depending on the distribution of the data. Regression analysis was used to identify potential risk factors associated with poor outcomes.

### Patient and public involvement

Although there was no direct involvement of patients and public in this study the Meningitis Research Foundation, a key advocacy group for patients are represented in the authorship of the original guidelines and will be key in the dissemination of the results and the subsequent call to improve practice. Preliminary results have been shared

with the Meningitis Research Foundation and some of their members.

## RESULTS

1471 patients from 64 hospitals throughout the UK and Ireland took part (see online supplemental appendix 1). The hospitals ranged in size from small district generals to large teaching hospitals. The mean number of beds was 846 (range 230–2000). The hospitals who took part in England comprised 45% of the total acute bed base in England, (42 612/94 827).[14] Females accounted for 57% (n=838) and the median age was 34 years (IQR 26, 49). Confirmed viral meningitis occurred in 615 (42%) and 303 had confirmed bacterial meningitis (21%). More than one-third of patients (n=553) fulfilled the case definition (box 1) but had no confirmed microbiological diagnosis and were therefore categorised as meningitis of unknown aetiology. Using the criteria proposed by Spanos et al,[15] 56 of those without a confirmed aetiology could be assumed to have bacterial meningitis. S. pneumoniae and N. meningitidis were the most common bacterial pathogens, where a cause was found, accounting for 172 (57%) and 76 (25%) of cases, respectively. H. influenzae (serotypes unknown) was found in 14 cases. Enteroviruses were the most common viral pathogens occurring in 429 (69%) of all confirmed viral meningitis. Herpes simplex virus-2 was the second most common viral pathogen detected in 97

(16%) of viral cases. Baseline demographics and clinical characteristics are shown in table 1.

Adherence to specific standards of good practice is shown in table 2. None were adhered to 100% of the time. Two-thirds of the standards (n=20) had ≤50% adherence.

Overall, in-hospital mortality was low (48/1471 (3%)). The mortality was higher in bacterial meningitis (28/302, 13%), and pneumococcal meningitis in particular (28/172, 16%). Mortality in viral meningitis was 0.3% (2/615) and 1.5% (8/548) in those with meningitis of unknown aetiology. Just over half (157) of those with confirmed bacterial meningitis required admission to an intensive care unit (ICU).

### Use of diagnostics

A few patients, 42, did not have an LP, of whom 26 (62%) had no contraindication (as specified in the 2016 joint specialties guidelines and shown in box 2). Five had meningococcal sepsis without clinical evidence of meningitis. The remaining 37 had clinical symptoms of meningism as well as a positive blood culture (n=35, 83%) and/or a positive blood PCR (n=16, 38%) for either S. pneumoniae (n=23, 55%), N. meningitidis (n=18, 43%) or L. monocytogenes (n=1, 2%).

Contraindications for immediate LP were uncommon and occurred in 299 (20%) patients. Glasgow Coma Score (GCS)≤12 was the most common contraindication for immediate LP reported in 143 (10%), followed by

**Table 1** Baseline demographics, timing of key investigations and clinical outcomes of 1471 adults presenting with suspected meningitis

| | Total cohort N (%) | Bacterial meningitis N (%) | Viral meningitis N (%) | Other* N (%) | P value† |
|---|---|---|---|---|---|
| N | 1471 (100) | 303 (21) | 615 (42) | 553 (38) | – |
| Median age (IQR) | 34 (26–49) | 54 (36–65) | 31 (25–37) | 34 (26–48) | <0.001 |
| Male | 625 (43) | 173 (57) | 214 (35) | 238 (43) | <0.001 |
| In patient mortality | 48 (3) | 38 (13) | 2 (0.3) | 8 (1.4) | <0.001 |
| Intensive care unit admission | 192 (13) | 157 (53) | 4 (0.7) | 31 (6) | <0.001 |
| Median admission GCS (IQR) | 15 (14–15) | 13 (9–15) | 15 (15–15) | 15 (15–15) | <0.001 |
| Median time (hours) from admission to first antibiotics (IQR) | 2.7 (0.9–8.3) | 1.5 (0.4–5.3) | 3.2 (1.3–8.3) | 3.3. (1–12.5) | <0.001 |
| Median time (hours) from admission to blood cultures (IQR) | 1 (0.3–4) | 0.7 (0.2–2.4) | 1 (0.3–3.7) | 1.4 (0.3–6.1) | 0.003 |
| CT of the head prior to LP | 1094 (94) | 207 (93) | 459 (94) | 428 (95) | 0.55 |
| Median time (hours) from admission to LP (IQR) | 16.4 (7.9–26.7) | 14.8 (7.7–29.8) | 14.3 (7.5–22.6) | 20 (8.8–35.8) | <0.001 |
| Adjunctive dexamethasone | 300 (21) | 150 (50) | 69 (11) | 81 (15) | <0.001 |
| Median CSF leucocyte count (IQR) | 140 (44–399) | 930 (235.5–3062.5) | 122 (48–276) | 85 (26.8–250.3) | <0.001 |
| Median CSF protein (IQR) | 0.68 (0.46–1.21) | 3.25 (1.4–5.8) | 0.63 (0.45–0.9) | 0.6 (0.4–1.0) | <0.001 |
| Median CSF glucose (IQR) | 3.2 (2.8–3.7) | 2.1 (0.95–3.45) | 3.2 (2.9–3.6) | 3.3 (3.0–3.8) | <0.001 |

*Other meningitis category included all patients without a confirmed bacterial or viral pathogen.
†For continuous variables, the Kruskal-Wallis test was used to compare medians across groups, and for categorical variables χ² tests were used.
CSF, cerebrospinal fluid; GCS, Glasgow Coma Score; LP, lumbar puncture.

**Table 2** Adherence to audit standards*

| Immediate management | Number achieved standard/total number of patients analysed | % of total | Number achieved standard/total number of patients evaluable† | % of number evaluable |
|---|---|---|---|---|
| 1. The patient's conscious level should be documented using the Glasgow Coma Scale | 1283/1471 | 87% | 1283/1448 | 89% |
| 2. Blood cultures should be taken as soon as possible and within 1 hour of arrival at hospital | 326/1471‡ | 22% | 326/767§ | 42% |
| 3. LP should be performed within 1 hour of arrival at hospital provided that it is safe to do so | 8/1471¶ | 0.5% | 8/1379** | 0.6% |
| 4. Antibiotic treatment should be commenced within the first hour | 207/1471†† | 14% | 207/1083‡‡ | 19% |
| 5. Patients with meningitis and meningococcal sepsis should be cared for with the input of an infection specialist such as a microbiologist or a physician with training in infectious diseases and/or microbiology | 1148/1471§§ | 78% | 1148/1464 | 78% |
| **Investigations** | | | | |
| 6. Blood culture should be sent | 977/1471 | 66% | 977/1469 | 67% |
| 7. Blood pneumococcal PCR should be sent | 211/1471 | 14% | 211/1460 | 14% |
| 8. Blood meningococcal PCR should be sent | 232/1471 | 16% | 232/1461 | 16% |
| 9. CSF opening pressure should be documented | 655/1428¶¶ | 46% | 655/1361[a] | 48% |
| 10. CSF glucose with concurrent plasma glucose should be sent | 607/1428¶¶ | 43% | 607/1415 | 43% |
| 11. CSF protein should be sent | 1358/1428¶¶ | 95% | 1358/1420 | 96% |
| 12. Microscopy of the CSF should take place within 2 hours of the lumbar puncture | 596/1428¶¶ | 42% | 596/1203[b] | 50% |
| 13. CSF for pneumococcal PCR should be sent in all cases of suspected bacterial meningitis | 412/1428¶¶ | 29% | 412/1418 | 29% |
| 14. CSF for meningococcal PCR should be sent in all cases of suspected bacterial meningitis | 434/1428¶¶ | 30% | 434/1418 | 31% |
| 15. A swab of the posterior nasopharyngeal wall should be obtained as soon as possible, and sent for meningococcal culture, in all cases of suspected meningococcal meningitis/sepsis | 54/1471 | 4% | 54/1463[c] | 4% |
| 16. All patients with meningitis should have an HIV test | 646/1471 | 44% | 646/1459[d] | 44% |
| **Treatment** | | | | |
| 17. All patients with suspected meningitis or meningococcal sepsis should be given ceftriaxone or cefotaxime | 1039/1471[e] | 71% | 1039/1423[f] | 73% |
| 18. If the patient has, within the last 6 months, been to a country where penicillin resistant pneumococci are prevalent, intravenous vancomycin 15–20 mg/kg should be added 12-hourly (or 600 mg rifampicin 12-hourly intravenous or orally)[g] | See footnote | | | |
| 19. Those aged 60 or over should receive 2 g intravenous ampicillin/amoxicillin 4-hourly in addition to a cephalosporin (1B) | 55/233 | 24% | 55/197[h] | 28% |
| 20. Immunocompromised patients (including diabetics and those with a history of alcohol misuse) should receive 2 g intravenous ampicillin/amoxicillin 4-hourly in addition to a cephalosporin | 26/115[i] | 23% | 26/99[j] | 26% |
| 21. If there is a clear history of anaphylaxis to penicillins or cephalosporins give intravenous chloramphenicol 25 mg/kg 6-hourly | 14/37 | 38% | 14/30[k] | 47% |
| 22. If *Streptococcus pneumoniae* is identified continue with intravenous benzylpenicillin 2.4 g 4-hourly, 2 g ceftriaxone intravenous 12-hourly or 2 g cefotaxime intravenous 6-hourly | 114/172 | 66% | 114/145[l] | 79% |
| 23. If number of meningitidis is identified 2 g ceftriaxone intravenous 12-hourly, 2 g cefotaxime intravenous 6-hourly or 2.4 benzylpenicillin intravenous 4-hourly may be given as an alternative | 52/76 | 68% | 52/68[m] | 76% |
| 24. If the patient is not treated with ceftriaxone (in meningococcal disease), a single dose of 500 mg ciprofloxacin orally should also be given | 0/2 | 0% | 0/2 | 0% |
| 25. If *Listeria monocytogenes* is identified Give 2 g ampicillin/amoxicillin intravenous 4-hourly and continue for at least 21 days. Cotrimoxazole 10–20 mg/kg or chloramphenicol are alternatives in cases of anaphylaxis to beta lactams | 4/7 | 57% | 4/6 | 67%[n] |
| 26. If *Haemophilus influenzae* is identified continue 2 g ceftriaxone intravenous 12-hourly or 2 g cefotaxime intravenous 6-hourly for 10 days | 9/14 | 64% | 9/13 | 69%[o] |
| 27. 10 mg dexamethasone intravenous 6-hourly should be started on admission, either shortly before or simultaneously with antibiotics | 67/1471 | 5% | 67/1435[p] | 5% |
| 28. If pneumococcal meningitis is confirmed dexamethasone should be continued for 4 days | 34/172[q] | 20% | 34/158[r] | 22% |

Continued

**Table 2** Continued

| Immediate management | Number achieved standard/total number of patients analysed | % of total | Number achieved standard/total number of patients evaluable† | % of number evaluable |
|---|---|---|---|---|
| **Critical care** | | | | |
| 29. The following patients should be transferred to critical care—those with a rapidly evolving rash, those with a GCS of 12 or less and those with uncontrolled seizures | 151/203ˢ | 74% | 151/203 | 74% |
| **Notification** | | | | |
| 30. All cases of meningitis (regardless of aetiology) should be notified to the relevant public health authority | 236/1471 | 16% | 236/1465 | 16% |

*Only those audit standards that could be measured from the data collected.
†Excludes those where there were missing data and/or where not relevant.
‡Only 977 patients had blood cultures taken.
§Excluding those who did not have blood cultures taken and where data were missing.
¶1428 patients had an LP.
**Excludes those who did not have an LP and where data were not available.
††82 patients had data consistent with having antibiotics prior to admission, this might be due to confusion about whether admission meant admission to the emergency department or admission to a ward, or it may represent data entry error therefore, these figures are not included.
‡‡388 patients did not receive any antibiotics at all.
§§310 (21%) of patients were admitted under an infection specialist, all others received consulting advice only.
¶¶43 people did not have an LP.
ᵃMissing data on 67.
ᵇ43 had no LP, 97 missing data, 128 time of microscopy was before or at the same time as the LP.
ᶜPerformed in 15/76 (20%) of proven meningococcal cases.
ᵈ9 known HIV positive and 3 missing data.
ᵉ285 patients were not given any antibiotics at all.
ᶠ48 patients who were definitely given antibiotics had missing data on which antibiotics they were given.
ᵍUsing mainland Europe data only and with reference to ECDC data—101 patients were documented to have travelled to a mainland European country within the previous 6 months. Travel history was not documented at all in 822 cases (56%). Of the 101 patients who had travelled to mainland Europe 54 (54%) had been to a country with a rate of penicillin resistant pneumococci of >5% (2017 data). 5/52 had no antibiotics. 0/47 had antibiotics to cover for penicillin resistant pneumococci.
ʰ233 patients were aged over 60 but only 207 received antibiotics. Missing data for 10, 108 received amoxicillin at some point but only 55 received the correct dose.
ⁱNot including those ≥60.
ʲ15 did not received any antibiotics and missing data on 1.
ᵏ7 patients had no antibiotics at all.
ˡ27 patients had insufficient antibiotic data.
ᵐ8 patient had insufficient antibiotic data.
ⁿ1 patient had insufficient antibiotic data.
ᵒInsufficient antibiotic data on 1 person.
ᵖMissing data on 36—11 on whether dexamethasone was received or not, 21 on the dose given and 4 on the timing.
q Only 18 were given the correct dose (10 mg). Some received dexamethasone for longer than 4 days.
ʳMissing data on 14 individuals.
ˢ7/11 patient with progressing rash, 131/176 patients with GCS <13 and 13/16 patients with uncontrolled seizures.
CSF, cerebrospinal fluid; GCS, Glasgow Coma Score; LP, lumbar puncture.

focal neurological signs in 38 (3%). A further 70 (7%) had other indications to delay LP. Neuroimaging prior to LP happened in 1094 of 1158 patients (94%), 911 (83%) of whom had no guideline-specified indication. Neuroimaging was performed a median of 11 hours post arrival at hospital (IQR 4–21). Median time from admission to LP was 16.5 hours (IQR 8–27). Only 6 patients had an LP within 1 hour of arrival at hospital and only 326 (26%) within 8 hours.

Median time from LP to CSF microscopy was 2 hours (IQR 1.1–3.2). Time from LP to CSF analysis was significantly quicker when performed at on-site laboratories when compared with centralised laboratory processing (median 1.65 hours (IQR 1.0–2.8) compared with 2.95 hours (IQR 2.0–3.8) p<0.001).

> **Box 2 Indications for neuroimaging before lumbar puncture in suspected meningitis**
>
> ⇒ Focal neurological signs
> ⇒ Presence of papilloedema
> ⇒ Continuous or uncontrolled seizures
> ⇒ Glasgow Coma Score≤12

Fewer than one-third of patients had pneumococcal (412, 28%) and meningococcal PCR (434, 29.5%) performed on their CSF. Pneumococcal PCR was done on blood in 211 (14%) patients, and meningococcal PCR in 232 (16%). Overall, 646 patients (44%) patients had a documented HIV test. Four of these were positive—two of whom had pneumococcal meningitis, one of whom had enteroviral meningitis and one had meningitis of unknown aetiology. Nine patients were previously known to be HIV positive.

Blood cultures were taken from 66% (n=977) of patients with 45% (n=438) having them taken within 1 hour of arrival at hospital.

### Treatment

Overall, 285 patients (19%) did not receive antibiotics, most of whom had either viral meningitis (163) or lymphocytic meningitis with no aetiology identified (105). The remaining 1186 patients received at least one dose of antibiotics. The median time from hospital admission to first dose of antibiotics was 3.2 hours (IQR 1.3, 9.2). Among the patients who received antibiotics the antimicrobials were commenced within an hour of arrival at hospital for approximately one-fifth of patients (207/1000). In

confirmed bacterial meningitis cases, 92 patients (36%) received antibiotics within an hour of arrival.

Adherence with guideline specified empirical antibiotic regimens was good with 912 (80%) receiving a third-generation cephalosporin. Data are missing on antibiotic type for 47 patients. Of the 197 patients aged 60 years and over who received antibiotics, 108 (55%) received ampicillin or amoxicillin; only 55 (28%) of those had the correct dose and dosing frequency as recommended for *L. monocytogenes* meningitis. Similarly, only 36 (31%) of the immunocompromised patients, who were aged under 60, (n=115) received any ampicillin or amoxicillin for anti-*Listeria* cover. Online supplemental table 1 shows details regarding risk factors for *Listeria*.

Only 300 patients (20%) received adjunctive steroids as recommended. Steroids were given more frequently in patient with confirmed bacterial meningitis in 150 (50%) cases. In patients with pneumococcal meningitis, 97 patients (57%) received steroids.

### Clinical outcomes

On multivariate analysis, having a confirmed diagnosis of bacterial meningitis was strongly associated with in-hospital mortality. Adjusting for age and sex, confirmed bacterial meningitis was associated with 26 times the odds of in-hospital mortality compared with those with other forms of meningitis (adjusted OR (aOR) 25.9, 95% CI 5.93 to 113.0), including those with no aetiology identified.

In patients with confirmed bacterial meningitis, on univariate analyses, in-hospital mortality was associated with a positive blood culture (crude OR (cOR) 2.21, 95% CI 1.04 to 4.67); GCS≤13 (cOR 3.24, 95% CI 1.39 to 7.52); confirmed *S. pneumoniae* meningitis (cOR 2.37, 95% CI 1.10 to 5.11); and ICU admission (cOR 4.81, 95% CI 1.99 to 11.60). These associations remained despite multivariate adjustment for age and sex (table 3).

The analysis was also conducted using only data from those who had had an LP (online supplemental table 2). The association between a positive blood culture and mortality was lost. The association between confirmed pneumococcal aetiology and mortality was approaching statistical significance and the association of ICU admission was maintained.

On both univariate and multivariate analyses (adjusted for age and sex), in patients with confirmed bacterial meningitis, the administration of dexamethasone was associated with a reduction in in-hospital mortality (aOR 0.57, 95% CI 0.28 to 1.17, p 0.12). When this analysis was restricted to include only those with confirmed *S. pneumoniae* meningitis, those who received dexamethasone had a reduced odds of in-hospital mortality (aOR 0.47, 95% CI 0.20 to 1.10, p 0.08). Neither association reached statistical significance. This analysis was also performed including the patients assumed to have bacterial meningitis according to the Spanos criteria (online supplemental table 3).

## DISCUSSION

This large national study evaluated clinical management of adults with community-acquired meningitis throughout the UK and Ireland. Current practice falls short of the recommendations in the 2016 UK guidelines.[6] This is a concern for all patients but is of a particular worry in bacterial meningitis. The management of bacterial meningitis is time critical.[4 16] Delays in receiving antibiotics and having an LP, the unnecessary use of brain imaging, a lack of appropriate antibiotics in those at risk of *Listeria* and the low rate of steroid administration are areas for significant improvement.

Most patients were given antibiotics prior to LP. Even taking this into consideration, the median door to antibiotic time was over 3 hours. The optimal timing of antibiotics in bacterial meningitis is not known precisely but we do know that delays lead to increased mortality.[4 5 16] A delay of over 3 hours has been associated with a 14-fold increase risk of death.[16]

Delays in obtaining CSF are associated with a reduction in pathogen detection, increased exposure to unnecessary anti-infectives, prolonged hospital stays and increased mortality.[4 6 17] In most cases, brain imaging is not indicated in adults with suspected community-acquired meningitis;[4] however, in our cohort, a significant number of patients had unnecessary scans. Although complications following LP are rare,[18 19] there may be an unfounded fear of cerebral herniation following LP, even in those with no clinical features of brain shift, which is leading to excessive use of imaging.[20] Education programmes, along with quality improvement measures, are essential to reduce the potentially harmful overuse of neuroloimaging. Additionally, it is essential that we optimise care pathways to ensure that clinicians have the time, space and equipment required to performed LPs in a timely and safe manner.[3 21]

CSF culture positivity rates decline substantially when LP is delayed.[3 17] PCR can detect bacterial DNA in CSF for several days after antibiotics have been administered. In the UK, half of meningococcal disease is diagnosed on PCR alone.[22] It is alarming that PCR was used, in our cohort, as a diagnostic modality in so few patients. Meningitis-specific investigation order-sets using electronic ordering, and/or reflex laboratory testing to increase use of molecular diagnostics should be considered to reduce opportunities for missed microbiological diagnoses. There is the potential for increased use of rapid technologies that can be used on site with minimal technical skill required.[23] Having rapid tests on site has been shown to reduce bed days with significant cost-savings.[24] Further research evaluating rapid diagnostic tests in other types of meningitis with clinically relevant outcomes is needed. We also need to increase the offer of HIV testing in patients with meningitis, as less than half the patients had a documented HIV test. Incident HIV diagnoses were

**Table 3** Multivariate analysis of the association between baseline covariates and in-hospital mortality in 303 patients with confirmed bacterial meningitis using logistic regression modelling

| Baseline covariate | N | In-hospital mortality N (%)* | Crude OR for in-hospital mortality (95% CI) | P value | Adjusted OR for in-hospital mortality (95% CI)† | P value‡ |
|---|---|---|---|---|---|---|
| Sex | | | | | | |
| Male | 173 | 26 (15.1) | 1 | | | |
| Female | 130 | 12 (9.23) | 0.57 (0.27 to 1.18) | 0.13 | | |
| Age group | | | | | | |
| ≤18 years | 18 | 0 (0) | | | | |
| 19–59 years | 159 | 18 (11.3) | 1 | | | |
| ≥60 years | 126 | 20 (16.0) | 1.49 (0.75 to 2.96) | 0.25 | | |
| Blood culture positive | | | | | | |
| No | 137 | 11 (8.09) | 1 | | 1 | |
| Yes | 166 | 27 (16.3) | 2.21 (1.04 to 4.67) | 0.03 | 1.87 (0.87 to 4.01) | 0.10 |
| GCS≤13§ | | | | | | |
| No | 124 | 8 (6.45) | 1 | | 1 | |
| Yes | 148 | 27 (18.2) | 3.24 (1.39 to 7.52) | 0.004 | 2.90 (1.26 to 6.71) | 0.008 |
| IV dexamethasone given¶ | | | | | | |
| No | 149 | 23 (15.4) | 1 | | 1 | |
| Yes | 150 | 14 (9.40) | 0.57 (0.27 to 1.16) | 0.11 | 0.57 (0.28 to 1.17) | 0.12 |
| Intravenous dexamethasone given if *Streptococcus pneumoniae*** | | | | | | |
| No | 73 | 16 (21.9) | 1 | | 1 | |
| Yes | 97 | 11 (11.5) | 0.46 (0.20 to 1.08) | 0.07 | 0.47 (0.20 to 1.10) | 0.08 |
| Final diagnosis *S. pneumoniae* | | | | | | |
| No | 131 | 10 (7.63) | 1 | | 1 | |
| Yes | 172 | 28 (16.4) | 2.37 (1.10 to 5.11) | 0.02 | 2.08 (0.96 to 4.48) | 0.05 |
| ICU admission†† | | | | | | |
| No | 144 | 7 (4.86) | 1 | | 1 | |
| Yes | 157 | 31 (19.7) | 4.81 (1.99 to 11.60) | <0.001 | 4.28 (1.81 to 10.1) | <0.001 |

*7/11 patient with progressing rash, 131/176 patients with GCS <13 and 13/16 patients with uncontrolled seizures.
†Adjusted for sex and age group.
‡P value from Likelihood ratio test comparing models with and without primary exposure variable.
§31/303 (10%) participants did not have a GCS recorded.
¶4/303 (1%) participants had missing data on intravenous dexamethasone administration.
**2/172 (1%) participants with confirmed *S. pneumoniae* meningitis had missing data on intravenous dexamethasone administration.
††1/303 (0.3%) participants had missing data on ICU admission.
GCS, Glasgow Coma Score.

made in our cohort among patients presenting with bacterial, viral and unknown cause meningitis.

There is good evidence that corticosteroids reduce mortality in pneumococcal meningitis with no clinically significant increase in adverse events in other causes of meningitis.[25] Empirical steroids should be given for all adults with suspected bacterial meningitis. In our study, we saw a reduction in mortality in patients with pneumococcal meningitis who were given steroids, while this survival benefit did not reach statistical significance, this was likely due to a type two error and the small sample of confirmed pneumococcal meningitis cases. It is of concern that well-evidenced, well-established therapies known to improve outcome, including mortality, are only being given to just over half those who might benefit. A protocolised, goal-directed bundle, including the use of corticosteroids and appropriate antibiotics, warrants evaluation in the UK. There were clear differences between centres in our study with one centre administering steroids to 26/42 (63%) of their patients and another giving them to none. It is possible that those centres that adhered to the recommendation to give steroids may also have adhered to other aspects of the guidelines more often as well, contributing to improved outcomes.

Although this is a large multinational study, there are limitations. NHS trusts self-selected themselves for inclusion, we cannot rule out any significant differences with trusts that did not. However, 64 hospitals were included with good representation throughout the nations of the UK (and Ireland). We do not think any potential selection bias limits the generalisability of our findings. We used

well-established, published case definitions of meningitis to minimise information bias; however, misclassification of cases remains possible especially in the cases without a confirmed microbiological diagnosis. Our case definitions allowed us to include anyone suspected of having meningitis (of any cause) as objectively it is often difficult to differentiate between viral and bacterial meningitis at the point of initial assessment. However, it is possible that there may have been differences in presentation between those with confirmed bacterial meningitis, those with confirmed viral meningitis and those with no confirmed aetiology that meant they were managed in different ways. This study was not powered to look at the differences between all the different aetiologies. Finally, because this was a retrospective study, our analysis may have been subject to errors resulting from recall bias and missing data. A prospective national study would have been challenging to execute and it is likely that there would have been ascertainment bias in time and geography. We therefore believe that, due to the large sample size along with the use of electronic hospital coding and laboratory data to ascertain cases, the risk of recall bias is low, and our retrospective data is representative of practice within the UK.

There is a clear need to better understand the suboptimal guideline adherence reported here. Although there has been research regarding primary care practice, there has not been any evaluation of exactly where delays occur and what the barriers are to achieving good practice in secondary care.[26 27] A small questionnaire-based study identified the inability to find correct equipment, lack of time and/or paucity of appropriately trained staff as potential barriers to performing timely LP for the investigation of neurological infections.[21]

Non-meningitis-specific research evaluating barriers and facilitators to adhering to clinical guidelines, report a lack of awareness or familiarity with the guidelines, as well as disagreement with the content may both be important.[28] External barriers such as equipment and staffing were also identified which agrees with the limited research that there is in neurological infections. There is observational evidence from other countries of improvements in practice and outcome following implementation of guidelines.[12 29]

The patient journey in the UK normally starts with being admitted via an emergency department or acute medical unit where clinicians may not be as familiar with the guidelines and evidence as specialists. There is some evidence, both within meningitis and other infectious diseases that management is improved by being looked after by a specialist. There is an expert recommendation within the current UK guidelines that patients with meningitis should be looked after with input of an infection specialist.

In conclusion, this is, to our knowledge, the largest UK study of adult patients with meningitis. Awareness of practice guidelines for relatively rare acute medical conditions such as meningitis is low and this study has demonstrated that despite clear, freely accessible guidelines, clinical care is not in line with evidence-based recommendations. There is considerable room for improvement. While we recognise that guidelines do not improve practice on their own, we do recommend that the findings from this study are strongly considered in the development of the new National Institute for Clinical Excellence (NICE) guideline on meningitis currently being developed, which for the first time, will include guidance for adult patients as well as children. Given the widespread adoption of NICE endorsed guidelines and quality standards to improve the quality of clinical practice in the UK, we anticipate that a NICE guideline will improve awareness and uptake of good practice in the short term. In addition to education, which has limited impact on changing behaviour, UK hospitals should use quality improvement methods to improve management of patients with suspected meningitis. Good qualitative research to identify what the barriers to implementing the guidelines should also be done.

We suggest a national strategic improvement plan should focus on the following key areas: timely use of diagnostics; appropriate antibiotics in at risk populations and the use of adjunctive steroids. The integrated use of electronic systems to standardise optimal use of diagnostics, and management bundles may offer additional opportunities to improve outcomes. Each site that has been involved in this study has been asked to implement site-specific changes and re-evaluate for any improvements in practice.

**Author affiliations**
[1]Research Department of Infection, Division of Infection and Immunity, University College London, London, UK
[2]Microbiology, Wirral University Teaching Hospital NHS Foundation Trust, Wirral, UK
[3]Institute of Infection, Veterinary and Ecological sciences, University of Liverpool, Liverpool, UK
[4]Tropical Infectious Diseases Unit, Liverpool University Hospitals NHS Foundation Trust, Liverpool, UK
[5]Clinical Research Department, London School of Hygiene and Tropical Medicine, London, UK
[6]University of Leeds, Leeds, UK
[7]National Student Association of Medical Research, Leeds, UK
[8]Neurology, The Walton Centre NHS Foundation Centre, Liverpool, UK
[9]Infectious Diseases and Medical Microbiology, Leeds Teaching Hospitals NHS Trust, Leeds, UK

**Collaborators** The online supplemental appendix 1 includes a list of other contributors in the National Audit of Meningitis Management (NAMM) group. National Audit of Meningitis Management (NAMM) group: Amy Chue, Ed Moran, Karishma Gokani, Joseph Thompson, Katherine Ajdukiewicz, Victoria Ward, Lucinda Barrett, Frances Edwards, Adam Usher, Mairi McLeod, Ramandeep Singh, Su su Htwe, Benedict Rogers, Grace Duane, Martin Wiselka, Nicholas Wong, Elen Vink, Jennifer Poyner, Jenni Crane, Ollie Lloyd, Emma Chisholm, Ildiko Kustos, Ruth McEwen, Sam Sutton, Lewis Jones, Robert Tilley, M. Estee Torok, Isobel Ramsay, Monica Ivan, Joshua York, Jennifer Ansett, Maithili Varadarajan, Celestine Eshiwe, Amanda Fife, Stephanie Harris, Ryan Jayesinghe, Priya Sekhon, James Cruise, Susan Larkin, Shivani Kanabar, Ernest Mutengesa, Mirella Ling, Christopher Green, Martin Williams, Matthew Stevens, David Griffith, Naomi Bulteel, Charlotte Milne, Jayanta Sarma, Aline Wilson, John Shone, Lynn Urquhart, Sahar Eldirdiri, Alison Muir, Leila White, Jody Aberdein, Phillip Simpson, Hnin Hay Mar, John Bowen, Keying Tan, Eint Shwe Zin thein, Mahmoud Aziz, Anthony Cadwgan, Brendan Davies,

Daniel White, Natasha Weston, Salman Zeb, Angela Houston, Imogen Fordham, Terry John Evans, Louise Wootton, David Turner, Iona Willingham, Aimee Johnson, Nimal Wickramasinghe, Ashley Horsley, Eamonn Trainor, Olivier Gaillemin, Andrew Rosser, Nicholas J Norton, Iain Crossingham, Katie Cheung, Megan Duxbury, Ashutosh Deshpande, Emilie Bellhouse, Kamaljit Khalsa, Helena Brezovjakova, Emma McLean, Tanmay, Kanitkar, Nicholas Davies, Alexsander Dawidziuk, Joanna Allen, Razan Saman, Sarah Kelly, Hugh Adler, Elshadai Ejere, Aarti Shah, Yiwen Soo, Wendy Beadles, Heather Sturgeon, Brodie Cameron, Ben Tomlinson, David Chadwick, Claire McGoldrick, Katie McDowell, Alastair Miller, Clive Graham, Mpho Molosiwa, Ewan Hunter, Ruth Owen, Katherine FlackAdrian Kennedy, Amy Robinson, Phoebe Cross, Fay Perry, Vithusha Inpadhas, Ali Khan, Sarathy Selvam, Vhairi Bateman, Jeremy Wong, Henry Wu, Monika Pasztor, Trupti Patel, Ajanthiha Karunakaran, Basma Soliman, Hassan Paraiso, Mairi McLeod, Su su Htwe, Anna Smith, Andrew Blanshard, Harish Reddy, Avneet Shahi, Helen Chesterfield, Oliver Bannister, Ben Schroeder, Ken Woodhouse, Jan Coebergh, Viva Levee, Eavan Muldoon, Rhea O'regan, Tee Keat Teoh, Sathyavani Subbarao, Simon Tiberi, Caryn RosmarinLucy Bell, Jonathan Lambourne, Emma McGuire, Robert Serafino, Anna Goodman, Ishaan Bhide, Karanjeet Sagoo, Mark Melzer, Maria Krutikov, Indran Balakrishnan, Susan Hopkins, Tim Jones, Kajal Patel, Barzo Faris, Graeme Calver, Ricky Singh, Hazel Sanghvi, Mohamed Eltayeb, Rathur Haris.

**Contributors** JE: Methodology, data collection and curation, formal analysis, investigation, writing—original draft preparation. DH: Methodology including pilot data, data collection, reviewing and approving final draft. SD: Methodology including development of original audit tool and guidelines, data collection, reviewing and approving final draft. AC: Methodology, reviewing and approving final draft. EM: Methodology, data collection, reviewing and approving final draft. TS: Methodology including development of original guidelines and audit tool, reviewing and approving final draft. RSH: Conceptualisation, methodology, supervision, writing—review and editing. FM: Conceptualisation, methodology, data collection and curation, investigation, formal analysis, writing—original draft preparation. Responsible for overall content as guarantor. The corresponding author attests that all listed authors meet authorship criteria and that no others meeting the criteria have been omitted. Please see online supplemental appendix 1 for list of other contributors in NAMM.

**Funding** The authors have not declared a specific grant for this research from any funding agency in the public, commercial or not-for-profit sectors.

**Competing interests** RSH is an NIHR Senior Investigator. The findings and the views expressed are those of the authors and not necessarily those of the NIHR. TS is supported by the National Institute for Health Research (NIHR) Health Protection Research Unit in Emerging and Zoonotic Infections (grant no. NIHR200907), NIHR Global Health Research Group on Brain Infections (no. 17/63/110) and the UK Medical Research Council's Global Effort on COVID-19 Programme (MR/V033441/1).

**Patient and public involvement** Patients and/or the public were involved in the design, or conduct, or reporting, or dissemination plans of this research. Refer to the Methods section for further details.

**Patient consent for publication** Not applicable.

**Ethics approval** As all data were anonymised individual patient consent and ethical approval was not required. The study was registered with each site's clinical governance department in line with local procedure.

**Provenance and peer review** Not commissioned; externally peer reviewed.

**Data availability statement** Data are available upon reasonable request. Data can be made available to other researchers on reasonable request to the authors.

**ORCID iDs**
Robert S Heyderman http://orcid.org/0000-0003-4573-449X
Fiona McGill http://orcid.org/0000-0002-0903-9046

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
