## [Reviewer comments · BMJ Open]

ARTICLE DETAILS

TITLE (PROVISIONAL)	Clinical management of community-acquired meningitis in adults in the United Kingdom and Ireland in 2017: a retrospective cohort study on behalf of the National Infection Trainees Collaborative for Audit and Research (NITCAR)
AUTHORS	Ellis, Jayne; Harvey, David; Defres, Sylviane; Chandna, Arjun; MacLachlan, Eloisa; Solomon, Tom; Heyderman, Robert; McGill, Fiona; study group, NAMM

VERSION 1 – REVIEW

REVIEWER	Chaudhuri, Abhijit Essex Centre for Neurological Sciences, Neurology
REVIEW RETURNED	11-Oct-2021

GENERAL COMMENTS	There are several shortcomings of this submitted work which is weakened by its retrospective nature and recall bias. The first is the assumption that the management of meningitis must adhere to the UK Joint Specialist Societies Guideline which I doubt is either a common or a necessary knowledge among physicians involved in acute intake of suspected meningitis. The authors instead should have set the minimum or core standards of expected clinical management, rather than trying to assess adherence to 29 standards (I cannot think of 29 standards to comply with management of any acute clinical condition). Second, the study is not that large. Fewer than 300 patients had acute bacterial meningitis (ABM) from 64 hospitals (<5/hospital/year), and they should have focused on confirmed or probable cases of ABM. Third, a small number of their patients (18) were in paediatric age group, which should have been excluded. Fourth, the presentation of clinical data of ABM patients is poor. It is not clear how many of 115 immunocompromised patients had ABM; the nature of immunosuppression and breakdown of etiological cause of ABM among these patients is not readily available. Fifth, the authors should have provided a simple table of LP data in ABM patients. If the information was available, they would be surprised to find that how infrequently corresponding plasma glucose is sent for comparison with CSF glucose, or how rarely opening pressure is recorded after LP. Sixth, there is no information about the specialties or grades of physicians treating ABM. In most cases, regrettably, the management is not consultant-led at the point of intake. Sixth, there is no information on non-fatal complications of ABM patients, e.g., seizures, hydrocephalus, stroke etc. Seventh, there is no tabulated information on admission GCS and GOS/Modified Rankin score in ABM patients stratified by age, immunological status and causative bacterial pathogen.
--

	I think the authors' suggestion of a guideline from NICE solving the problem of substandard clinical management of ABM in this country is an over-simplistic assumption. There is an appalling lack of proper clinical skills and knowledge in the management of neurological infectious diseases among average UK trainees and physicians. The manuscript barely skims the surface of this problem which affects the overall quality of care in ABM in this country, and does not offer a possible solution.
--	---

REVIEWER	Guillem, Lluïsa Hospital Universitario de Bellvitge, Infectious Diseases Department
REVIEW RETURNED	26-Nov-2021

GENERAL COMMENTS	I miss the description of the hospitals which agreed to enrol the study, just because they are a large number they could be not representative enough, a minimal description of how many of them are large/medium/small and a comparison it with all the NHS. I think some corrections should be done in the results. In the summary table describe adherence to the guidelines by items, but the “denominator” are all the cases with no missing data which is not accurate. The recommendations of the guidelines are made for the bacterial meningitis not the viral. If viral meningitis is suspected no antibiotics should be given neither PCR of pneumococcal or meningococcal. Also, in the statistical analysis, I think the main contribution of this article could be mortality assessed by guideline standard accomplished or not: for exemple by blood culture performed, not by result because of course is known positive blood culture will become a worst outcome. The same by “microscopy in 2 hours” ... This result would reinforce the importance of following the guidelines standards. To note it is true some of them are already shown for exemple dexhametasone use. I think the delay on PL or diagnose could not only be due to CT scan, because “time until suspicion” and waiting time until visit in the Emergency Department could also be the cause. With this I mean it is probable all viral meningitis had less “acute and severe” symptoms, reason for having that much delay. I should say that in other countries, symptoms during > 24h or immunosuppression are indicative of CT scan before LP due to the risk of brain abscess. Maybe the analysis could be assessed with just the bacterial meningitis cases. Also: 5: I can't find this explication note number below the supplementary table
---

VERSION 1 – AUTHOR RESPONSE

Reviewer: 1

Dr. Abhijit Chaudhuri, Essex Centre for Neurological Sciences

Comments to the Author:

There are several shortcomings of this submitted work which is weakened by its retrospective nature and recall bias. The first is the assumption that the management of meningitis must adhere to the UK

Joint Specialist Societies Guideline which I doubt is either a common or a necessary knowledge among physicians involved in acute intake of suspected meningitis. The authors instead should have set the minimum or core standards of expected clinical management, rather than trying to assess adherence to 29 standards (I cannot think of 29 standards to comply with management of any acute clinical condition).

We thank the reviewer for his comment regarding the retrospective nature of the study and the potential for recall bias. However, a prospective national study across so many sites would have been challenging to execute and it is likely that there would have been an ascertainment bias both in time and geography. We therefore maintain that, due the large sample size (1471 cases) and the use of electronic hospital coding and laboratory data to ascertain cases, the risk of recall bias is low, and our retrospective data is still representative of practice within the UK. We have emphasised this in the limitations section of the manuscript.

Our aim was to assess adherence to good practice in relation to the management of suspected meningitis, not the knowledge of specific guidelines. Ideally, acute physicians should know where to look for guidance on best practice. The reviewer rightly highlights that awareness of practice guidelines for relatively rare acute medical conditions such as meningitis is frequently low and we have highlighted this in the manuscript.

To make an assessment of good practice, we have used the most recent, most comprehensive consensus UK guidelines (NICE guidance for the management of meningitis in adults is currently being formulated). Many acute physicians in training consult “UpToDate” which although written by experts from the USA, provide a link to the UK Joint Specialist Societies Guidelines: https://www.uptodate.com/contents/society-guideline-links-bacterial-meningitis-in-adults?search=meningitis&topicRef=1287&source=see_link. The UK Joint specialist societies guidelines were a collaborative effort by all specialist societies that are likely to manage meningitis in adults along with patient group representatives. The societies involved in writing the guidelines were the Society for Acute Medicine, the Intensive Care Society, The Association of British Neurologists, Public Health England (as was), Meningitis Research Foundation and the British Infection Association.

Setting our own standards that differed from this expert consensus group would not have been appropriate. Many NICE guidelines will have more than 29 recommendations. We chose clinical, operational and laboratory standards from the UK Joint Specialist Societies guidelines that could be objectively measured with the data collected.

Second, the study is not that large. Fewer than 300 patients had acute bacterial meningitis (ABM) from 64 hospitals (<5/hospital/year), and they should have focused on confirmed or probable cases of ABM.

The clinical problem that we have assessed in this study is the investigation and management of suspected community-acquired meningitis in adults. As many elements of good practice in relation to meningitis are required before the final diagnosis is known (a frequent situation in acute medical emergencies), it is appropriate to have included all patients with meningitis, rather than just those with proven bacterial meningitis. Indeed, by taking this approach our data is comparable with the existing meningitis clinical practice literature.

In response to the reviewer's comment we have therefore added "suspected" to the title and where appropriate in the text. This remains the largest study of the management of suspected meningitis in the UK to date.

Third, a small number of their patients (18) were in paediatric age group, which should have been excluded.

The inclusion criteria in the pre-specified study protocol included patients aged 16 years or over which is often the age cut off for acute medical services within the NHS (many UK hospitals do not have specific wards for adolescents). The 18 patients that the reviewer refers to were between the age of 16 and 18 years. To now exclude these patients from our analysis post-hoc could be criticised and given that this small group represents <2% of the total cohort, is unlikely to introduce any bias.

Fourth, the presentation of clinical data of ABM patients is poor. It is not clear how many of 115 immunocompromised patients had ABM; the nature of immunosuppression and breakdown of etiological cause of ABM among these patients is not readily available.

The aetiological agents of confirmed bacterial meningitis are given for the main three pathogens (accounting for 262/302 cases (86%)) in the first paragraph of the results section. We now have added in the other bacterial causes. However, it should be emphasised that this is not primarily a study of the aetiology of ABM. The relevance of gathering data on immunocompromise was to assess the practice point of whether patients with risk factors were given appropriate antibiotics to cover for *Listeria monocytogenes* meningitis. Data was gathered on whether patients had certain risk factors which might make *Listeria monocytogenes* more common (age >60, immunocompromise, diabetes and excess alcohol use). In response to the reviewer's concerns, we have included a supplementary table showing the breakdown of each type of immunocompromise within each aetiological strata. There were a total of 115 patients with risk factors for *Listeria monocytogenes* meningitis. We have changed the manuscript to make this clearer.

Fifth, the authors should have provided a simple table of LP data in ABM patients. If the information was available, they would be surprised to find that how infrequently corresponding plasma glucose is sent for comparison with CSF glucose, or how rarely opening pressure is recorded after LP.

We thank the reviewer for this suggestion, we agree with the reviewer that corresponding plasma glucose and opening pressure are frequently not recorded – again highlighting the need to improve practice. Out of the patients who had a lumbar puncture 607 (43%) had a paired plasma glucose done within 4 hours of the CSF one. 655 patients (46%) had an opening pressure performed. We have added some simple LP investigation data to table 1.

Sixth, there is no information about the specialties or grades of physicians treating ABM. In most cases, regrettably, the management is not consultant-led at the point of intake.

We agree that senior and specialist input should be sought early. To address the reviewer's concern, we did record the date and time the patient was first seen by a 'senior decision maker' – defined as ST3 or above. The median time from admission to senior review was 3.9 hours (IQR 1.4,10). We have included this data within the manuscript. The supplementary table gives details on the number of patients who had the input of an infection specialist – namely an infectious diseases doctor or a microbiologist - at some point during their admission – 78%.

Sixth, there is no information on non-fatal complications of ABM patients, e.g., seizures, hydrocephalus, stroke etc. Seventh, there is no tabulated information on admission GCS and GOS/Modified Rankin score in ABM patients stratified by age, immunological status and causative bacterial pathogen.

As stated, the aim of this study was to assess practice not outcome and associated risk factors. ITU admissions and death are reported in table 1. We have added the GCS data into table 1. Glasgow Outcome and Modified Rankin Scores were not collected.

I think the authors' suggestion of a guideline from NICE solving the problem of substandard clinical management of ABM in this country is an over-simplistic assumption. There is an appalling lack of proper clinical skills and knowledge in the management of neurological infectious diseases among average UK trainees and physicians. The manuscript barely skims the surface of this problem which affects the overall quality of care in ABM in this country, and does not offer a possible solution.

We agree with the reviewer that a NICE guideline will not solve the problem and this has been emphasised in the manuscript. Given the widespread adoption of NICE endorsed guidelines to improve the quality of clinical practice, we anticipate that a NICE meningitis guideline will improve awareness and uptake of good practice in the short term, particularly if the guideline is highlighted in electronic resources such as UpToDate. To achieve a more sustainable change in practice, we have included several other suggestions in our concluding paragraph including local quality improvement strategies, a national strategic improvement plan and integrated use of electronic systems.

We would like to emphasise that one of the first steps in improving practice is understanding where we are now. This paper ensures the current, inadequate standard of practice is highlighted. By being published in BMJ Open, a journal aimed at the generalist rather than the specialist, ensures a wide audience amongst physicians who are likely to encounter this rare, but deadly, disease.

Dr. Lluïsa Guillem, Hospital Universitario de Bellvitge

Comments to the Author:

I miss the description of the hospitals which agreed to enrol the study, just because they are a large number they could be not representative enough, a minimal description of how many of them are large/medium/small and a comparison it with all the NHS.

Thank you for suggesting we include this useful data. We have included a more detailed description at the beginning of the results section. The hospitals ranged in size from small district generals to larger teaching hospitals. The mean number of beds was 846 (range 230,2000). The hospitals who took part in England comprised 45% of the total acute bed base in England, (42,612/94,827).

I think some corrections should be done in the results. In the summary table describe adherence to the guidelines by items, but the “denominator” are all the cases with no missing data which is not accurate. The recommendations of the guidelines are made for the bacterial meningitis not the viral. If viral meningitis is suspected no antibiotics should be given neither PCR of pneumococcal or meningococcal.

We thank the reviewer for her comments. We agree that not considering the missing data would be inaccurate. We included two different denominators in this table – one includes all cases, the other only includes those where data was available. We apologise if this was not clear and this has now been corrected. Additionally we have now placed this table into the main manuscript to make the data more accessible.

The best practice that we have evaluated is for all patients with suspected meningitis (see comments to reviewer 1). As differentiating between viral or bacterial meningitis is rarely straightforward clinically at the time of giving antibiotics or doing the LP, it is recommended that ALL patients with suspected meningitis should be given antibiotics within one hour and should have PCR tests requested. Of course, if a viral aetiology is identified the antibiotics should be discontinued. We have added in an explanatory sentence at the start of the methods section to make the rationale for including all patients clearer.

Also, in the statistical analysis, I think the main contribution of this article could be mortality assessed by guideline standard accomplished or not: for example by blood culture performed, not by result because of course is known positive blood culture will become a worst outcome. The same by “microscopy in 2 hours” ... This result would reinforce the importance of following the guidelines standards. To note it is true some of them are already shown for example dexamethasone use.

We thank the reviewer for this suggestion but would like to emphasise that that the primary aim of the study was to assess adherence to best practice. Our study was not designed to assess the association between guideline standards and mortality. Such an analysis would be underpowered and may suffer from misleading results arising from multiple testing.

I think the delay on PL or diagnose could not only be due to CT scan, because “time until suspicion” and waiting time until visit in the Emergency Department could also be the cause. With this I mean it is probable all viral meningitis had less “acute and severe” symptoms, reason for having that much delay. I should say that in other countries, symptoms during > 24h or immunosuppression are indicative of CT scan before LP due to the risk of brain abscess. Maybe the analysis could be assessed with just the bacterial meningitis cases.

We agree with the reviewer that time until ‘suspicion of meningitis’ is important however it was not possible to collect this data retrospectively. We have not suggested that the delay in LP or diagnosis is specifically due to the CT scan because, as the reviewer states there are other factors to consider. We acknowledge that different countries may have different indication for CT scanning but emphasise that there is similar guidance in the US and Europe.

Also:

5: I can’t find this explication note number below the supplementary table

Thank you for pointing this out. We have corrected this error.

VERSION 2 – REVIEW

REVIEWER	Chaudhuri, Abhijit Essex Centre for Neurological Sciences, Neurology
REVIEW RETURNED	27-Jan-2022

GENERAL COMMENTS	The revised manuscript adds little clarity to the nature of the retrospective study but there are multiple confounding issues. The head of the revised manuscript is suspected meningitis which includes non-infective lymphocytic meningitis, seroconversion meningitis and unconfirmed meningitis of presumed infective origin. A retrospective case series of "suspected" acute bacterial meningitis would gain less traction than a prospective case series of suspected meningitis. Authors should have focused primarily to the analysis of confirmed cases of bacterial meningitis in terms of adherence to treatment standards and clinical outcomes which I believe is the key research question they are seeking to answer. I am surprised that they chose to exclude tuberculous meningitis in this series, which remains a common presentation of subacute community acquired meningitis in younger patients, at least in Greater London area, with devastating outcome due to delayed diagnosis and poor early management. Authors' definitions of meningitis and encephalitis (box 1) would also not pass closer scrutiny. To define a cut off of 4 WCC in CSF as abnormal is not quite right; manual of RCP UK accepts a CSF white cell count of up to 5 and lymphocytes up to 3 to be normal. I hope the cut-off of CSF cell count was not retrospectively set up
--

	for case numbers: would the authors please confirm? All definitions of meningitis and encephalitis in box 1 appear rather arbitrary and not referenced. It is also not clear if autoimmune encephalitis was excluded reliably in their patients. I have to reiterate that for real purpose of their work, authors should focus their retrospective analysis to confirmed cases of acute bacterial meningitis and revise the manuscript accordingly. A statistical comparison of outcome analysis for confirmed vs suspected cases of acute bacterial meningitis would be useful.
REVIEWER	Guillem, Lluïsa Hospital Universitario de Bellvitge, Infectious Diseases Department
REVIEW RETURNED	14-Jan-2022
GENERAL COMMENTS	Good revision!

VERSION 2 – AUTHOR RESPONSE

Reviewer: 2

Dr. Lluïsa Guillem, Hospital Universitario de Bellvitge

Comments to the Author:

Good revision!

- ***Thank you for the complimentary review.***

Reviewer: 1

Dr. Abhijit Chaudhuri, Essex Centre for Neurological Sciences

Comments to the Author:

The revised manuscript adds little clarity to the nature of the retrospective study but there are multiple confounding issues. The head of the revised manuscript is suspected meningitis which includes non-infective lymphocytic meningitis, seroconversion meningitis and unconfirmed meningitis of presumed infective origin.

- **We are sorry that in contrast to the other reviewer, this reviewer is still not clear about the nature of the study. The aim of this retrospective study was to assess clinical practice in hospitals in the UK and Ireland in the management of patients with suspected bacterial meningitis and compared to best practice in the *UK guidelines on the diagnosis and management of acute meningitis* to inform clinical practice improvements, and future guidelines. This aim, and the nature of the study is clearly laid out in the background and methods section of the paper. See lines 105-108.**
- **We disagree with the reviewer's implication that there are multiple unidentified confounding issues that impact on the validity of this study. We maintain that the process outcomes that we have measured such timeliness of blood cultures, lumbar puncture, first dose of antibiotics are unlikely to have been markedly distorted by an association between the population selected and another unmeasured factor. We have**

also reported an analysis of key associations with outcome such as pneumococcal aetiology, admission to intensive care, initial Glasgow Coma Scale score and dexamethasone therapy. We have addressed the potential confounders in the limitations section and have not overstated causality in our discussion. Indeed, we recognise the diagnostic challenges in confirming acute meningitis, and the need for broad and sensitive clinical criteria to ensure cases are not missed, we have discussed the limitations as raised by reviewer one - including diagnostic misclassification of cases - in detail within the discussion. See lines 295-301.

- The reviewer is correct that “suspected bacterial meningitis” may include non-infective lymphocytic meningitis, HIV seroconversion meningitis and unconfirmed meningitis of presumed infective origin. The first two entities are relatively rare in routine acute medical practice and unconfirmed meningitis of presumed infective origin is entirely within the scope of the *UK guidelines on the diagnosis and management of acute meningitis* and the NICE Guideline for Meningitis (bacterial) and meningococcal disease: recognition, diagnosis and management currently being formulated, which also includes adults (<https://www.nice.org.uk/guidance/qid-ng10149/documents/final-scope>). It is therefore entirely justified to include all patients presenting with suspected bacterial meningitis in our analysis of the clinical pathway. Clinical symptoms and signs cannot reliably distinguish bacterial from viral meningitis or other differential diagnoses, therefore investigations, and indeed empirical antibiotic treatment must be commenced, on the basis of suspicion of bacterial meningitis while awaiting results of diagnostic procedures and well before the patient is known to have bacterial meningitis. We recognise that this will inevitably include patients that don't turn out to have confirmed bacterial meningitis but this does not impact on the findings. We have included more detail on this in the methods section – lines 118-123.
- It should also be noted the inclusion criteria for the study were clearly set out in a pre-defined protocol. It would be incorrect practice to change those inclusion criteria now.

A retrospective case series of "suspected" acute bacterial meningitis would gain less traction than a prospective case series of suspected meningitis. Authors should have focused primarily to the analysis of confirmed cases of bacterial meningitis in terms of adherence to treatment standards and clinical outcomes which I believe is the key research question they are seeking to answer.

- Given the size and the national reach of our retrospective study, we maintain that a prospective case series would have been logistically challenging and inevitably, due to unreported cases, smaller.
- The research question that we set out to answer was whether in current UK practice there is concordance with the UK guidelines in the management of adult patients with suspected community acquired meningitis and if there are areas for improvement. This is clearly stated in the manuscript (lines 105-108).
- Focussing on confirmed bacterial meningitis only would not have answered this question.

I am surprised that they chose to exclude tuberculous meningitis in this series, which remains a common presentation of subacute community acquired meningitis in younger patients, at least in Greater London area, with devastating outcome due to delayed diagnosis and poor early management.

- We agree that much could be done to improve the management of tuberculous meningitis (TBM) however, this was not the aim of this study. As outlined above the aim was to assess the management of patients who might be considered to have suspected community acquired bacterial meningitis. All current acute meningitis management guidelines in the UK, Europe and the USA exclude TBM. Patients with

TBM, as the reviewer states, would present in a different manner (subacute) and so would not be in the same clinical pathway. Furthermore, in 2020, there were 65 cases of TBM in England reported to UKHSA and therefore, to conduct a similar analysis for TBM would require data collected over multiple years which was outside the scope of this study.

Authors' definitions of meningitis and encephalitis (box 1) would also not pass closer scrutiny. To define a cut off of 4 WCC in CSF as abnormal is not quite right; manual of RCP UK accepts a CSF white cell count of up to 5 and lymphocytes up to 3 to be normal.

- **The Reviewer is incorrect in his conclusion. The cut off in our definitions is >4 white cells (which is the same as up to 5). This is consistent with the Standard in Microbiological Investigations published by the UK Health Security Agency (https://assets.publishing.service.gov.uk/government/uploads/system/uploads/attachment_data/file/618337/B_27i6.1.pdf - page 11).**

I hope the cut-off of CSF cell count was not retrospectively set up for case numbers: would the authors please confirm?

- **This is not the case - the CSF cell count cut off is well established as above. The inclusion criteria for the study were clearly set out in a pre-defined peer-reviewed protocol.**

All definitions of meningitis and encephalitis in box 1 appear rather arbitrary and not referenced. It is also not clear if autoimmune encephalitis was excluded reliably in their patients.

- **Definitions are consistent with definitions we have used before in several peer reviewed publications – McGill et al, Lancet 2016, McGill et al, Lancet Infectious Diseases 2018 and McGill et al, Journal of Infection 2022. We have now referenced the definitions (line 118). The Definitions box clearly states that encephalitis was excluded and this definition would include the majority of cases of autoimmune encephalitis. The reviewer is however correct that autoimmune encephalitis *may* manifest as suspected meningitis but this would be extremely rare and very unlikely to bias our findings.**

I have to reiterate that for real purpose of their work, authors should focus their retrospective analysis to confirmed cases of acute bacterial meningitis and revise the manuscript accordingly. A statistical comparison of outcome analysis for confirmed vs suspected cases of acute bacterial meningitis would be useful.

- **This is not the purpose of the study, and we argue that these analyses would do little to inform improvements in the clinical management of suspected meningitis cases in the UK and address a different question.**

VERSION 3 – REVIEW

REVIEWER	Brouwer, Matthijs University of Amsterdam, Department of Neurology
REVIEW RETURNED	04-Apr-2022

GENERAL COMMENTS	The authors have performed a retrospective study on the care for meningitis patients, both bacterial and viral, and those with no established cause. Although the study has several limitations, it shows that care for meningitis patients can improve in several ways and is thereby a useful message to clinicians. The study size is large as is the number of hospitals, which probably provides a representative sample of patients. Inclusion of the 43 patients without CSF examination is debatable, as we cannot be 100% sure they actually had meningitis. I would consider leaving them out. For the group with uncertain aetiology, it may be possible to filter out those with likely bacterial meningitis based on the criteria by Spanos et al (CSF glucose <1.9 mmol/L, CSF-blood glucose ratio <0.23, CSF protein >2.2 g/L, >2000 x 10(6)/L CSF leukocytes, or >1180 x 10(6)/L CSF polymorphonuclear leukocytes - individual predictors of bacterial infection with ≥99% certainty). This could be added to the results and does not have to result in regrouping of patients for all analysis. I was wondering whether there were substantial differences between centres in this study, for instance if there were centres in which nobody received dexamethasone and those in which (almost) everyone did. Is there a group of believers and non-believers in specific treatments or is it general lack of adherence to advised diagnostics and treatment of these patients? Dexamethasone treatment appears to be associated with better outcome, but potentially the hospitals in which dexamethasone is administered also adhere to other aspects of the guidelines better (e.g. earlier LP, adequate antibiotics) so it is difficult to analyse this part of treatment in isolation. This could be mentioned in the discussion. In my opinion previous RCTs, meta-analyses and implementation studies provide sufficient evidence for the efficacy of dexamethasone, so showing a non-significant effect of the treatment in a retrospective study may send a wrong signal. In the discussion the authors mostly describe why the separate guideline items are important to improve prognosis, but what is missing is discussion on why the adherence to the guideline is so poor. Are there any studies probing on the reasons to wait with the lumbar puncture or not to give dexamethasone? Are neurologists not willing to perform lumbar punctures in the night or weekend? The 2016 UK guidelines are quite similar to 2010 NICE guidelines, so it is unlikely that physicians were unfamiliar of most recommendations of the 2016 UK guidelines. A Swedish study suggested meningitis is better handled by ID physicians compared to no-ID physicians (PMID 25752223), although I realise the situation is quite different per country. Are there any thoughts on how to improve the situation in the UK/Ireland? A more general discussion on the effect of guidelines in meningitis could also include a prospective Dutch time series comparing treatment before and after introduction of guidelines (PMID 27484018), which showed that frequency of cranial imaging was not changed because of the guideline introduction, whereas antibiotic treatment and time to treatment was.
---

VERSION 3 – AUTHOR RESPONSE

The authors have performed a retrospective study on the care for meningitis patients, both bacterial and viral, and those with no established cause. Although the study has several limitations, it shows that care for meningitis patients can improve in several ways and is thereby a useful message to clinicians. The study size is large as is the number of hospitals, which probably provides a representative sample of patients.

1. Inclusion of the 43 patients without CSF examination is debatable, as we cannot be 100% sure they actually had meningitis. I would consider leaving them out.

We agree with the reviewer that it is indeed difficult to know if these patients had meningitis or not however we would nonetheless like to include them in the study as there are clinical scenarios of suspected meningitis in which it is not safe to do a lumbar puncture. These patients still need to be investigated +/- treated for meningitis.

This group of patients all had symptoms of meningitis as well as a significant blood culture or PCR for *Streptococcus pneumoniae* (n=18), *Neisseria meningitidis* (n=23) or *Listeria monocytogenes* (n=1). We have added in this information to make it clear all these patients had clinically likely meningitis. We have performed the multivariate analysis without this group and presented the results in a supplementary table.

2. For the group with uncertain aetiology, it may be possible to filter out those with likely bacterial meningitis based on the criteria by Spanos et al (CSF glucose <1.9 mmol/L, CSF-blood glucose ratio <0.23, CSF protein >2.2 g/L, >2000 x 10(6)/L CSF leukocytes, or >1180 x 10(6)/L CSF polymorphonuclear leukocytes - individual predictors of bacterial infection with ≥99% certainty). This could be added to the results and does not have to result in regrouping of patients for all analysis.

Thank you for this suggestion. We have performed the multivariate analysis including the extra 56 patients who using the Spanos criteria would have been classified as bacterial aetiology. This is presented in supplementary table 2.

3. I was wondering whether there were substantial differences between centres in this study, for instance if there were centres in which nobody received dexamethasone and those in which (almost) everyone did. Is there a group of believers and non-believers in specific treatments or is it general lack of adherence to advised diagnostics and treatment of these patients?

Thank you. This is an interesting point. There were some appreciable differences in adherence to dexamethasone provision across study sites for example 0% (0/10) of study participants at one site received dexamethasone, compared to 63% (26/41) of study participants at another. We have added this into the discussion but have not undertaken a comprehensive analysis as the sample size at each centre varied and the potential confounders are considerable.

4. Dexamethasone treatment appears to be associated with better outcome, but potentially the hospitals in which dexamethasone is administered also adhere to other aspects of the guidelines better (e.g. earlier LP, adequate antibiotics) so it is difficult analyse this part of treatment in isolation. This could be mentioned in the discussion. In my opinion previous RCTs, meta-analyses and implementation studies provide sufficient evidence for the efficacy of dexamethasone, so showing a non-significant effect of the treatment in a retrospective study may send a wrong signal.

Thank you for raising this point, we agree and hope that this study would encourage the use of corticosteroids more in the UK. We have adjusted the discussion to emphasise the already

established benefit of steroids and mention the point regarding the differential use of steroids and other management between centres.

5. In the discussion the authors mostly describe why the separate guideline items are important to improve prognosis, but what is missing is discussion on why the adherence to the guideline is so poor. Are there any studies probing on the reasons to wait with the lumbar puncture or not to give dexamethasone? Are neurologists not willing to perform lumbar punctures in the night or weekend? The 2016 UK guidelines are quite similar to 2010 NICE guidelines, so it is unlikely that physicians were unfamiliar of most recommendations of the 2016 UK guidelines. A Swedish study suggested meningitis is better handled by ID physicians compared to no-ID physicians (PMID 25752223), although I realise the situation is quite difference per country. Are there any thought on improve the situation in the UK/Ireland?

Thanks for raising this important issue. To understand why the specifics of the guideline were not followed good qualitative research is badly needed. As far as we are aware, while qualitative studies have been conducted in primary care, a qualitative study in secondary care has not been published with regard to meningitis specifically. There have been several reports into why guidelines in general aren't followed and what the specific barriers might be. We have added a section into the discussion on this. It should be noted that the 2010 NICE guidelines were for children only and did not include adults hence the clinicians that were managing the patients in our study would not have been familiar with them. The current ongoing revision of the 2010 NICE guidelines will include adults and so combine the 2016 UK guidelines and the 2010 ones.

We agree that the patient journey in the UK may contribute to the problem and have added a comment to that effect. Patients are often not initially seen by infection specialists or neurologists and it may be several days before they see a specialist. There was a significant trend in our study towards improved survival if participants were under the care of an infectious diseases team (cOR 0.24, 95% CI 0.07-0.78, p 0.02) however we felt there was too many confounders in this to be meaningful e.g. those patients on ITU would have a worse prognosis and not be under the care of an Infection specialist.

6. A more general discussion on the effect of guidelines in meningitis could also include a prospective Dutch time series comparing treatment before and after introduction of guidelines (PMID 27484018), which showed that frequency of cranial imaging was not changed because of the guideline introduction, whereas antibiotic treatment and time to treatment

Thank you – we have added in a section about the effect of guideline implementation in meningitis in both the Netherland and Sweden.

Edits:

*In the list of author names, please change "NAMM (national audit of meningitis management)" to "National Audit of Meningitis Management (NAMM)".

This is done.

*Please ensure periods are used at the end of all sections/sentences in the abstract and 'Strengths and limitations of this study' sections.

This is now done.

*Please update the 'Results' section of the abstract to include absolute numbers for all percentages reported. This should also be done in the main text (unless absolute numbers reported in a cited table).

This is now done.

*Please define aOR at first mention in the abstract and please define aOR and cOR at first mention in the main text. Any other abbreviations should also be defined at first mention in the main text.

This is done.

*Please revise the statement heading "Author contributions" to "Contributors" and please format "Competing interests" as an underlined heading to make clear this is a separate statement. Additionally, the sentence "Original data can be shared on request" should be deleted from the "Competing interests" statement, as this is covered elsewhere.

This is done.

*Regarding the list of non-author contributors ("List of contributors in NAMM"), the simplest solution would be to supply the list as an online appendix file (either in the same file as the supplementary table or a separate file) and to cite this list in the 'Contributors' statement. The tabular format does not work for inclusion in the statements at the end of the main manuscript.

Thanks. I have removed this list and submitted as a separate appendix file and cited in the contributors statement. Please note this list of contributors should be citable as authors as part of a group authorship policy if possible. Is this possible?

VERSION 4 – REVIEW

REVIEWER	Brouwer, Matthijs University of Amsterdam, Department of Neurology
REVIEW RETURNED	01-Jun-2022
GENERAL COMMENTS	The revisions have improved the paper and brought more balance in the discussion. I have no further comments